# The Peculiarities of Convective Heat Transfer in Melt of a Multiple-Electrode Arc Furnace

**Alexsey Kukharev [1,\*], Vyacheslav Bilousov [2,\*], Ecaterina Bilousov [2] and Vitaly Bondarenko [2]**

[1] State Educational Institution of the Higher Professional Education, Lugansk National University named after Vladimir Dahl, 92700 Lugansk city, Ukraine

[2] State Educational Institution of the Higher Professional Education, Donetsk National University, 21000 Donetsk city, Ukraine; s_bilousov@mail.ru (E.B.); vitbond@gmail.com (V.B.)

\* Correspondence: alex.kuharev@mail.ru (A.K.); v.v.bilousov@gmail.com (V.B.); Tel.: +380953028034 (A.K.); +380508054958 (V.B.)

**Abstract:** The modern direction of improving the technology of steel production in high-power arc furnaces is the intensification of magnetohydrodynamic effects for mixing the melt. In this article, a furnace design is proposed that contains three roof arc and three bottom electrodes, which provides the formation of additional eddy currents in the melt when the furnace is supplied with direct current or a low-frequency current. For a numerical study of the features of heat transfer in the melt of this furnace, a three-dimensional mathematical model of magnetohydrodynamic and thermal processes was used. The results were processed using the methods of visualization of vortex structures and the Richardson criterion. In an oven with a capacity of 180 tons at currents in the electrodes of 80 kA, the conditions for the interaction of electric vortex and thermogravitational convection were studied. Results showed that thermogravitational convection due to nonuniform heating of the melt led to a decrease in the size of the main electric vortex flow and the formation of an additional flow near the side walls of the furnace. The features of azimuthal flows formed in the areas of electric arcs and hearth electrodes were analyzed. Results showed that the multivortex structure of the flows that formed in the furnace allowed the volume of stagnant zones to be reduced and provided acceptable melt mixing conditions. The results can be used to improve the energy and structural parameters of three-electrode arc furnaces.

**Keywords:** arc furnace; magnetic hydrodynamics; electric vortex flows; heat-gravitational convection

## 1. Introduction

At present, high-power arc steel-making furnaces with melt bath capacity exceeding 100 tons and installed power exceeding 100 MVA with currents of 50–150 kA in the electrodes [1,2] may be referred to as a separate group among electrometallurgical units. Such furnaces are most commonly made as multiple-electrode ones to reduce loads on the electrodes and distribution of the input electric power [1–4].

An urgent task for furnaces with a large volume of metal is the effective mixing of the melt, which contributes to the intensification of heat transfer as well as its temperature and chemical homogenization. Along with the use of melt mixing by purging with various gases, electromagnetic mixing methods based on various magnetohydrodynamic effects are currently used [1–5].

At the same time, high melt temperatures, especially in the areas of electric arcs, as well as the chemical aggressiveness of the medium make experimental study of liquid metal flows in working furnaces practically impossible. Therefore, to improve mixing conditions, numerical modeling of processes seems to be the most promising method.

Numerical modeling in [6] showed that, in the absence of additional mixing means in the furnace bath, the main melt flows are localized in the regions of electric arcs, and stagnation zones are formed in the lower part of the bath and on its periphery, in which the homogenization processes will be significantly slowed down.

In [5], hydrodynamic and thermal processes in a three-electrode furnace with induction mixing using a bottom inductor were studied. The results of this work showed that electromagnetic mixing leads to a decrease in melting time and an increase in the heat transfer coefficient in the melt.

In [7–9], flows and heat transfer processes in direct current furnaces with one arc and one or two hearth electrodes were studied, in which the melt is mixed due to the so-called electric vortex flows, which are caused by the Lorentz forces as a result of the interaction of the electric current flowing in the melt current with its own magnetic field. However, under the conditions of reliable operation of graphitized electrodes, the power of such furnaces is limited to approximately 100 MVA. In [3], the structure of flows in a furnace with three arcs and one hearth electrode was studied.

In [4], flows in a furnace with two vaulted and four hearth electrodes were studied. Note that the furnace considered in [4] was installed at the Tokyo Steel plant and is one of the most powerful in the world (256 MVA). In these studies, the multivortex structure of flows was shown. In the region of electric arcs, downward flows of the melt are formed, and in the region of the hearth electrodes, upward flows are formed. In this case, thermogravitational convection affects the structure of flows depending on the temperature difference, direction, and intensity of eddy currents. However, these furnaces are designed more for DC power. The patent [10] was based on the task of intensifying magnetohydrodynamic effects and heat transfer in existing three-electrode furnaces by installing additional hearth electrodes, which are located in the centers of stagnant zones. Thus, in the furnace, the angle between the adjacent axes of the roof and bottom electrodes is $60°$. In addition, the power supply of the proposed furnace is proposed to be carried out from special converters that provide power supply with both direct current and low-frequency current (0.01–1 Hz) with adjustable parameters of amplitude, shape, phase, and frequency. According to [10], these parameters can be regulated independently for each arc electrode. Such regulation will provide effective correction of the electromagnetic field in the furnace melt and will thus allow the vortex electric flows and heat fluxes in the melt to be effectively controlled during the melting process.

This article explores the features of heat transfer in the melt of the proposed six-electrode furnace when it is supplied with direct current, taking into account the interaction of electric vortex and thermogravitational convection to further evaluate the feasibility of upgrading existing three-electrode arc furnaces. This study is a continuation of [11], which was devoted to numerical modeling of the electromagnetic field in this furnace.

## 2. Materials and Methods

The six-electrode arc furnace used in this research had a cylindro-spheroconical bath with a capacity of 180 tons. The bath geometric model and profile are shown in Figure 1. The basic geometric dimensions of the bath are also indicated there.

When constructing the model, all metals were assumed to be in a molten state (the heat final period); therefore, the lining impact on heat transfer was not taken into account. The electric arcs were represented in the model by homogeneous cylindrical conductors located in the centers of the cross section of the corresponding arc electrodes. Availability of concave menisci in the melt area under the electric arcs was also assumed [12].

The slag layer was not considered in the model, and the electrodes and the arcs were taken into account only when calculating the electromagnetic field [11]. The impact of the electric arcs was allowed for in the hydrodynamic and thermal calculations by introducing the corresponding boundary conditions in the meniscus area. This approach allowed us to simplify the hydrodynamic model and reduce the number of grid elements.

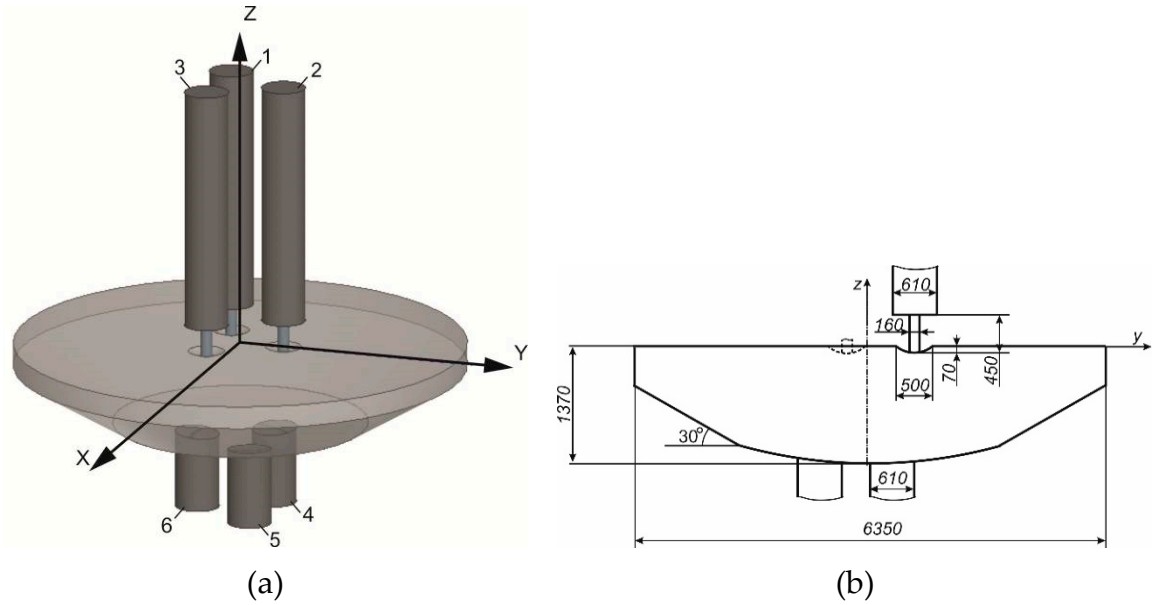

**Figure 1.** The geometric model of the furnace bath: (**a**) the general view; (**b**) the bath profile in the direction of the Y axis (Unit: mm). In (**a**), 1, 2, and 3 are roof arc electrodes, while 4, 5, and 6 are bottom electrodes.

When developing a mathematical model, the magnetohydrodynamic approximation was taken as the main assumption, where the processes under consideration were described by the system of equations of magnetic hydrodynamics [1,13]. In this case, chemical reactions were not taken into account, and the media within the highlighted zones were considered homogeneous, isotropic, and nonmagnetic. As all the metals in the final melting period were in a molten state and chemically active components were no longer introduced into the furnace, this assumption is acceptable in general.

As the magnetic Reynolds number was $Re_m < 1$, transfer of the magnetic field by the conductive fluid flow was not allowed for [1,13].

Due to various peculiarities of the process in the furnaces, heat transfer between the slag and the melt as well as influence of the fuel and oxygen burners were not considered at this stage. The Reynolds number for the melt flows was $Re = 10^4–10^5$, so we used the $k-\varepsilon$ model to describe turbulence. This model, being relatively simple, provides the velocity distributions that are closest to the experimental data [1,9,14]. Under the assumptions made, the basic equations of the model will look like this:

energy equation:

$$\rho C_p\left(\frac{\partial T}{\partial t}\right) = -\rho C_p\left(\vec{v}\cdot\nabla T\right) + \lambda_e\nabla^2 T + Q, \tag{1}$$

Navier–Stokes equation:

$$\rho\left[\frac{\partial \vec{v}}{\partial t} + \left(\vec{v}\cdot\nabla\right)\vec{v}\right] = -\nabla p + \eta_e\nabla^2\vec{v} + \rho\vec{g} + \vec{F}, \tag{2}$$

flow continuity equation:

$$\nabla\cdot\vec{v} = 0, \tag{3}$$

$k-\varepsilon$ turbulence model equations:

$$\frac{\partial(\rho k)}{\partial t} + \nabla\cdot\left(\rho\vec{v}k\right) = \nabla\cdot\left[\left(\eta + \frac{\eta_T}{\sigma_k}\right)\nabla k\right] + G_k + G_b - \rho\varepsilon, \tag{4}$$

$$\frac{\partial(\rho\varepsilon)}{\partial t} + \nabla\cdot\left(\rho\vec{v}\varepsilon\right) = \nabla\cdot\left[\left(\eta + \frac{\eta_T}{\sigma_\varepsilon}\right)\nabla\varepsilon\right] + C_{1\varepsilon}\frac{\varepsilon}{k}(G_k + C_{3\varepsilon}G_b) - C_{2\varepsilon}\rho\frac{\varepsilon^2}{k},\tag{5}$$

Kolmogorov–Prandtl equation:

$$\eta_T = \rho C_\mu\frac{k^2}{\varepsilon},\tag{6}$$

equation for turbulent thermal conductivity:

$$\lambda_T = \frac{C_p\eta_T}{Pr_T},\tag{7}$$

equation of state to describe the relationship between density and temperature:

$$\rho = \rho_0[1 - \beta(T - T_0)].\tag{8}$$

The model variables are as follows: $\vec{v}$ is the melt velocity in m/s; $p$ is pressure in Pa; $T$ is temperature in K; $\eta_e = \eta + \eta_T$ is effective dynamic viscosity coefficient in Pa·s; $\lambda_e = \lambda + \lambda_T$ is effective coefficient of the melt thermal conductivity in W/(m·K); $k$ is turbulent kinetic energy in m$^2$s$^2$; $\varepsilon$ is dissipation of the kinetic energy of turbulence in m$^2$/s$^3$; $G_k$ is specific turbulence generation from medium velocity gradients; $G_b$ is specific turbulence generation from the Archimedes force; $C_{3\varepsilon}$ is the coefficient depending on flow directions and gravitational force.

We determined the Lorentz forces in the melt $\vec{F} = \vec{J}\times\vec{B}$ and the density of Joule heating $Q = \left|\vec{J}\right|^2/\sigma$ based on the calculation of the electromagnetic field. The basic equations and the boundary conditions for the electromagnetic field are detailed in [11]. The furnace was simulated with currents of 80 kA in the electrodes.

We determined the characteristics of the electric arcs on the basis of a numerical solution of the Elenbaas–Heller equation [15]. To determine the heat flux from the arc to the meniscus region, a technique similar to that given in [16] was used.

The physical parameters of liquid steel, shown in Table 1, were used in the simulation. In the case of electric vortex melt flows, the regions of high temperatures are effectively transported deep into the bath and the temperature in the bath is averaged to values of 1950–2100 K, so changes in the thermal conductivity coefficient, dynamic viscosity coefficient, and heat capacity in this temperature range of approximately 15% are made [12,17]. Therefore, in this study, these parameters were assumed constant.

**Table 1.** The molten steel properties and the turbulence model constants.

| Parameter | Designation | Unit | Value |
|---|---|---|---|
| Density | $\rho_0$ | kg/m$^3$ | 6900 |
| Representative temperature | $T_0$ | K | 1900 |
| Dynamic viscosity coefficient | $\eta$ | Pa·S | 0.007 |
| Molecular thermal conductivity ratio | $\lambda$ | W/(m·K) | 35 |
| Heating capacity | $C_p$ | J/(kg·K) | 792 |
| Volumetric expansion coefficient | $\beta$ | K$^{-1}$ | 0.00014 |
| The eddy Prandtl number | $Pr_T$ | - | 0.85 |
|  | $C_{1\varepsilon}$ | - | 1.44 |
|  | $C_{2\varepsilon}$ | - | 1.92 |
| Constant turbulence models | $C_\mu$ | - | 0.09 |
|  | $\sigma_\varepsilon$ | - | 1.0 |
|  | $\sigma_k$ | - | 1.3 |

The constants of the turbulence model, given in Table 1, depend on the dynamics of the properties of liquids and were taken according to Launder [18].

The boundary conditions (BC) used are given in Table 2.

**Table 2.** Boundary conditions.

| Surface | Imposed Boundary Conditions (BC) | |
|---|---|---|
| | Hydrodynamic | Thermal |
| The bath lower and side walls | No-slip condition: $v = 0$ | Condition of the first kind: $T_b = 1900$ K |
| Meniscus areas of the electric arcs | No-slip condition: $v = 0$ | Condition of the second kind: $q_b = 6.43 \times 10^7$ W/m$^2$ [15] |
| The melt upper surface | No-slip condition: $v = 0$ | Thermal insulation condition: $q_b = 0$ |

Note that setting a constant temperature on the walls of the bath to some extent corresponds to the conditions of the final melting period in the furnace with a steady heat balance. Setting a constant heat flux density in the meniscus region, in our opinion, is quite acceptable due to the constant movement of the anode spots of the arc along the meniscus surface [12].

Note that a slag layer covering the melt upper surface is formed in the high-power arc furnaces during the smelting process; thus, the thermal insulation condition was applied on this surface. This boundary condition was also substantiated by Szekelly in [19] for a slag layer with a thickness of at least 6 inches. The thickness of the slag layer in powerful arc furnaces exceeds 6 inches.

## 3. Results and Discussion

The melt flows were studied at the first stage without taking into account the heat-gravitational convection. In this case, Equation (8) was not used, $\vec{g} = 0$ was assumed in Equation (2), and $G_b = 0$ was assumed in Equations (4) and (5).

The results of modeling the flows in the bath vertical section passing through the Y axis without consideration of the heat-gravitational convection are shown in Figure 2. The solid lines in the figure show the flow lines (velocity lines), and arrows indicate the flow directions.

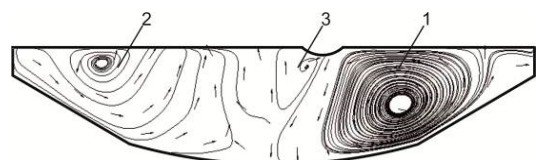

**Figure 2.** The trajectories of the melt flow in the bath vertical section along the Y axis, disregarding the heat-gravitational convection.

As can be seen in the figure, the melt flows had a multivortex structure. Powerful melt jets were formed in the area under the electric arcs. They were directed at an angle of approximately 20° (relative to the axes of the electric arcs) to the bottom of the furnace. The width of these jets in the middle part of the bath depth was about 800 mm. The melt jet maximum velocity reached 2.2 m/s. The melt spread out to the periphery with velocities of 0.2–0.5 m/s in the lower part of the bath. Then, the melt rose ($v < 0.2$ m/s) slowly to the surface in a section that was removed approximately 2500 mm from the bath axis and then flowed to the corresponding arc area, where it was again drawn by the electromagnetic forces deep into the bath. This flow is designated as the first circuit of the melt circulation in Figure 2.

The upward vortex flows were formed with a maximum speed of 0.5–0.7 m/s near the bottom electrodes in the areas between the electric arcs. The width of the upward flow jet in the middle of the bath was about 1200 mm. The second circulation circuit was clearly observed, where the melt moved up near the bath axis; on the surface, it moved slowly toward the periphery.

In addition, there was a small vortex moving in the opposite direction relative to the first circulation circuit in the area of the arc spot between the first and second circuits. The flow in this circuit was directed downward under the electric arc, but it turned upward under the action of the second vortex flow approximately in the middle of the bath depth. The melt movement on its surface was directed

from the bath center to the electric arc area with a maximum velocity of about 0.5–1 m/s directly under the meniscus of the arc.

Consideration of the heat-gravitational convection formed due to the uneven heating of the melt (Figure 3) in the model changed the structure of the flows slightly.

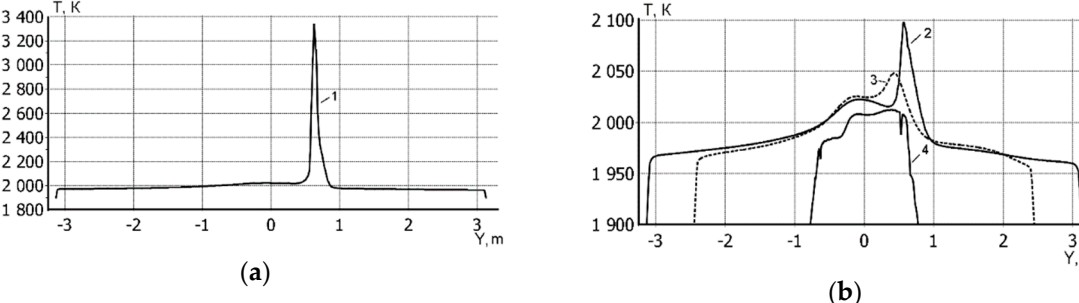

(**a**)                                        (**b**)

**Figure 3.** Graphs of the temperature distribution along the Y axis: (**a**) with Z = −70 mm (1); (**b**) with Z = −310 mm (2), with Z = −715 mm (3), and with Z = −1320 mm (4).

Generally, interaction of the electric vortex and the heat-gravitational convections can be explained using the scheme shown in Figure 4. It can be seen that the electric vortex and naturally convective flows were contradirectional in the area of the electric arcs, and the directions of the electric vortex and the heat-gravitational convections coincided in the area of the bottom electrodes.

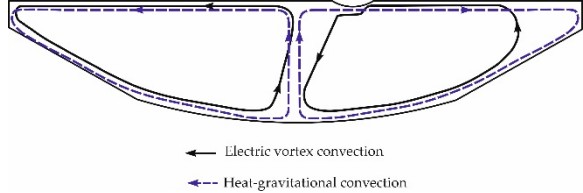

←— Electric vortex convection
←-- Heat-gravitational convection

**Figure 4.** Simplified scheme showing interaction of the electric vortex and heat-gravitational convections.

We also used the Richardson criterion to assess the heat-gravitational convection effect:

$$Ri = \frac{Gr}{Re^2}.$$

(9)

The graph of distribution of the local Richardson number along the bath diameter is shown in Figure 5. It can be seen that the values of the numbers were $Ri \approx 1$ near the bath axis. This suggests that the influence of the electric vortex convection was great, and the influence of the buoyancy forces was insignificant. $Ri$ was >3 near the side walls (the temperature difference was about 50 K), which was manifested in a decrease in the size of the first circulation circuit and the formation of the fourth circuit (Figure 6) in the structure of the flows in the vertical plane. In this case, the intensity of the flows in the second and third circuit rose.

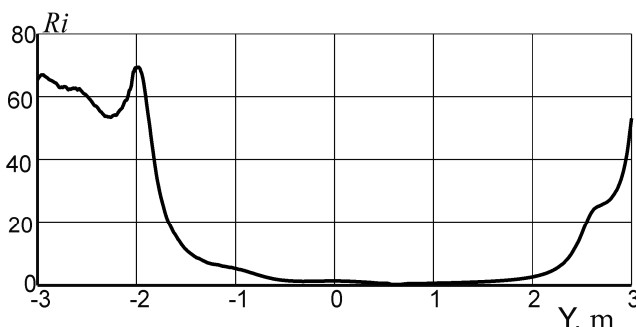

**Figure 5.** Distribution of the local Richardson number along the Y axis with Z = −75 mm.

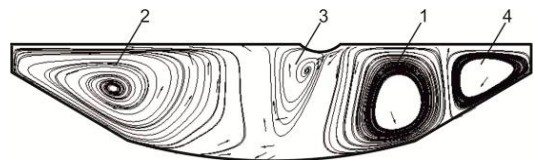

**Figure 6.** The trajectories of flows in the bath vertical section along the Y axis with allowance made for the heat-gravitational convection.

The numerical change in velocities with regard to the heat-gravitational convection is illustrated by the velocity distribution graphs (Figure 7). In these graphs, solid lines show the velocity profiles with consideration for the heat-gravitational convection, and dotted lines show them without consideration for the heat-gravitational convection.

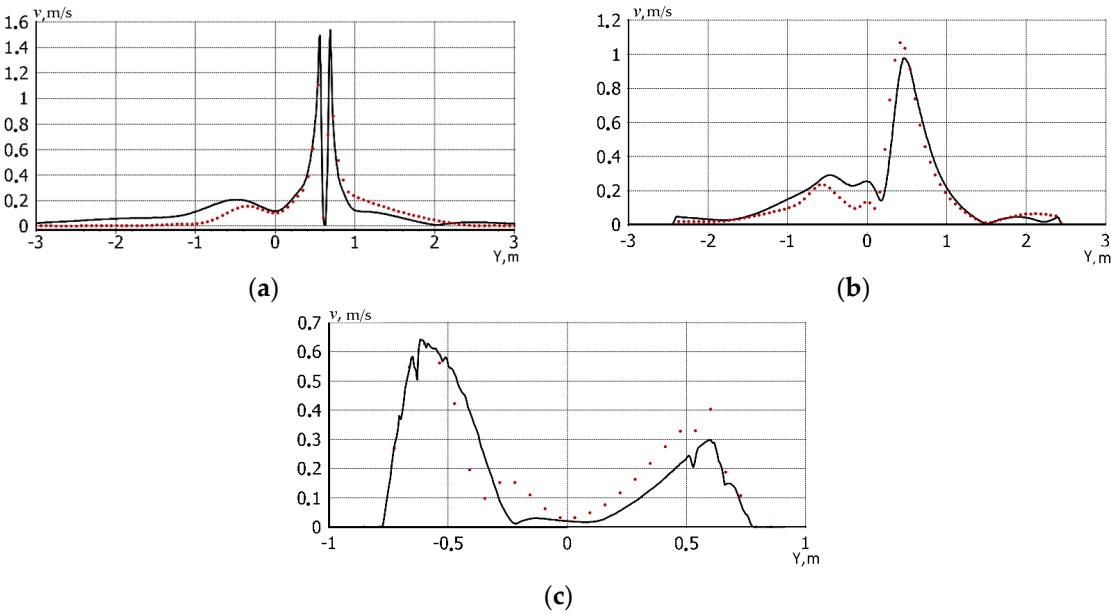

**Figure 7.** Graphs of the velocity distribution along the Y axis: (**a**) with Z = −70 mm; (**b**) with Z = −715 mm; (**c**) with Z = −1320 mm.

It can be seen in the graphs that the values of the downward flow velocities were lower in the areas under the electric arcs when the heat-gravitational convection was taken into account, and the velocities of the upward flows were enhanced in the bath axial area and in the area of the bottom electrodes. On average, the change in velocities was 15–25%.

It might be worth pointing out that the electric vortex convection interacted with the heat-gravitational convection in the bath horizontal sections too, thus resulting in azimuthal swirling

of the flows in the areas of the electric arcs and the bottom electrodes (Figure 8). These flows formed as three pairs of vortices in both areas. Moreover, the jets converged to the lines connecting the bath center and the axes of the electric arcs near the melt surface as well as to the lines connecting the bath center and the axes of the corresponding bottom electrodes in the area of the bottom electrodes.

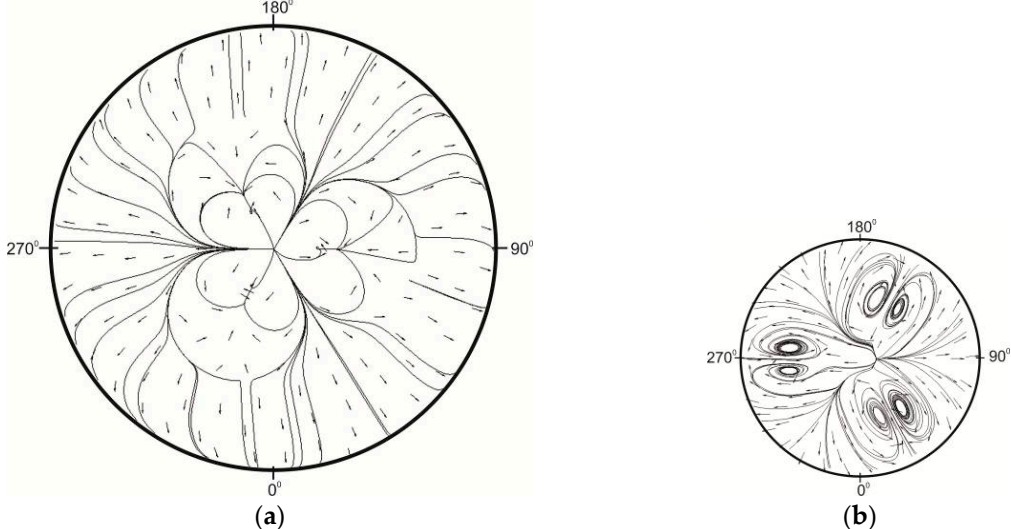

**Figure 8.** The trajectories of the melt flows in the bath horizontal sections, taking into account the heat-gravitational convection: (**a**) with Z = −70 mm; (**b**) with Z = −1320 mm.

It should be mentioned that the azimuthal swirling of the flows under the conditions of the electric vortex and heat-gravitational convections is quite a complicated phenomenon due to the appearance of azimuthal electromagnetic forces in the areas formed by the electric current lines converging to the electrodes, interaction of the bath intrinsic electromagnetic field with external magnetic fields, and availability of viscous friction effects [13].

We analyzed and compared the graphs of the azimuthal velocities and the electromagnetic forces near the melt surface and near the bottom electrodes. The graphs were plotted based on angular coordinate $\varphi$ along the electrode pitch circle diameter. The direction of the X axis was taken as origin of 0° (Figure 8). Near the melt surface, the maximum values of $v_\varphi$ were observed in the areas of the lateral surfaces of the spots of the electric arcs (Figure 9a), where $v_\varphi$ was about 1.1 m/s (in 83°, 95°, 203°, 215°, 323°, and 325°). The sign reversal for the azimuthal velocity corresponded to the flow divergence areas near coordinates 90°, 210°, and 330° and to the flow convergence areas near coordinates 30°, 150°, and 270°.

In the lower part of the bath (Figure 9 c), $v_\varphi$ reached 0.3 m/s near the lateral surfaces of the bottom electrodes (near coordinates 0°, 60°, 120°, 180°, 240°, 300°, and 360°). The flow divergence areas corresponded to coordinates 90°, 210°, 330°, and the flow convergence areas corresponded to coordinates 30°, 150°, 270°.

The approximate similarity of the distribution graphs for the azimuthal velocities and the electromagnetic forces, as well as coincidence of the coordinates of their extremes, suggests that the azimuthal flows were caused to a greater extent by the corresponding electromagnetic forces. In this case, the impact of the heat-gravitational convection on swirling of the flows near the melt surface was insignificant. In the area of the bottom electrodes, the effect of the naturally convective flows moving in the direction from the bath periphery to the center manifested itself in enhancement of the azimuthal velocity by approximately 20% as well as shift of the vortex centers to the bath center by approximately 500 mm.

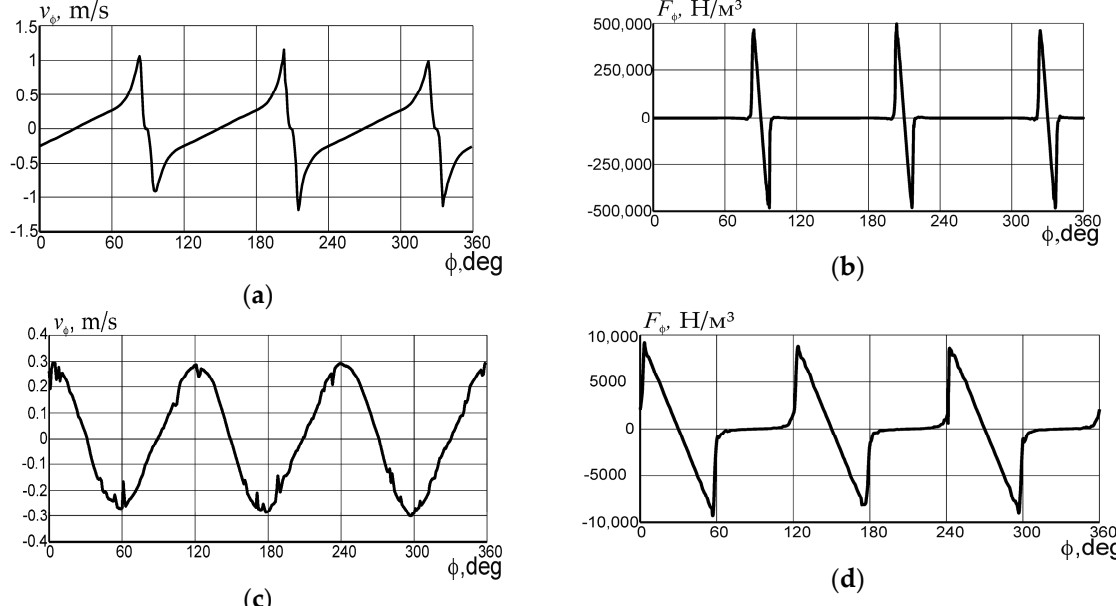

**Figure 9.** Comparison of the graphs of azimuthal velocity components and electromagnetic forces: (**a**,**b**) with Z = −70 mm; (**c**,**d**) with Z = −1320 mm.

In this study, the degree of melt mixing was estimated based on the volume of stagnant zones, that is, areas of the bath in which the flow rate of molten steel was lower than 0.01 m/s were defined as stagnation zones.

Then, the total relative volume of the stagnant zones, expressed through the volume of the melt in the bath, was as follows:

$$V'_d = \frac{V_d}{V_b} 100\% \tag{10}$$

where $V_d$ is the melt volume (in m$^3$) in which the velocity is less than 0.01 m/s; $V_b$ is the volume of the melt (in m$^3$) in the bath.

Note that according to the results of [5,6] and the results of our research in three-electrode furnaces without the use of additional mixing, the volume of stagnant zones can be more than 15%.

According to studies in a six-electrode furnace, the volume of stagnant zones was 1.008 m$^3$, which is 3.8% of the melt volume in the bath. These values are in good agreement with studies [20] carried out in a three-electrode furnace with bottom purging, in which the volume of stagnant zones varied from 0.49 to 1.31 m$^3$.

Thus, on the basis of research data, it is possible to make conclusions about acceptable mixing results when using the proposed circuitry and design solutions.

As a rule, laboratory installations with fusible metals or alloys are used to verify mathematical models of industrial furnaces that take into account the formation of electromagnetic forces in the melt [12–14]. The use of such metals or alloys makes it relatively easy to carry out high-precision measurements at temperatures up to 1000 K at almost any point in the melt. Relatively low radiation levels allow the use of simple and inexpensive thermocouples to measure temperatures.

To verify this mathematical model, we used the experimental data obtained at two electrode laboratory facilities of the Joint Institute for High Temperatures of the Russian Academy of Sciences [13] and the Massachusetts Institute of Technology [14]. To compare the results, we introduced geometric conditions and operating modes of these laboratory units into the model.

From [13], velocity profiles were used both in the axial and radial directions of a hemispherical bath filled with an indium–gallium–tin alloy and a current in the electrodes of 250 A. Note that these

measurements were carried out under specially recognized isothermal conditions at a temperature of 283 K.

As an example, Figure 10 shows a comparison of calculated and experimental data on a graph of the velocity distribution over the depth of the bath along its axis. As can be seen, Figure 10 confirms the satisfactory agreement of the model with the experimental data, while the average deviation of the experimental data from the calculated does not exceed 15%.

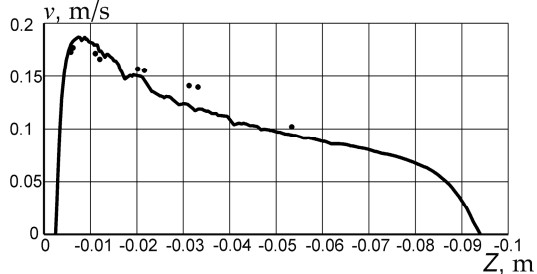

**Figure 10.** Graph of the velocity distribution over the bath depth along its axis (—- calculation; • experiment [13]).

Similar comparisons were made using experimental data in [14]. This laboratory setting was cylindrical and filled with a Wood alloy. The current in the electrodes was 500 A. Figure 11 shows a comparison of calculated and experimental data on a graph of the velocity distribution along the radius of the bath at Z = −221 mm. As can be seen, the satisfactory agreement between the model and experimental data is also confirmed but to a greater extent in the main flow region; the average deviation of the calculated and experimental data also does not exceed 15%. The convergence of the results was slightly worse near the side wall at $r = 0.07$ m, which can be explained by a simplified description of the processes in the wall region as well as possible measurement errors. These reasons were also discussed by the authors of [14].

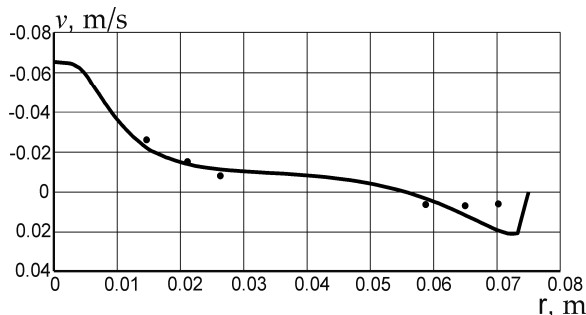

**Figure 11.** Graph of the velocity distribution along its radius at Z= −221 mm (—- calculation; • experiment [14]).

Due to the fact that the above experimental studies were carried out under isothermal conditions, to further verify the model, we developed a six-electrode laboratory setup that was geometrically similar to the furnace under study. Lead melt was used as a conductive fluid. The design of the installation and the measurement procedure are considered in detail in [21]. Figure 12 shows the results of comparing the calculated and experimental data on the graph of the distribution of the averaged temperature over the diameter of the bath near the surface of the melt. In this graph, the average deviation of the experimental and calculated data does not exceed 10%.

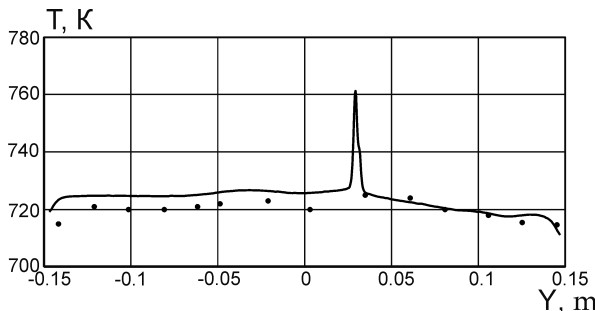

**Figure 12.** Graph of the temperature distribution along the Y axis (—— calculation; • experiment).

In addition, the presented structure of the azimuthal flows is similar to the structure obtained in a three-electrode, laboratory-scale plant with mercury [22]. We also note that the results obtained do not contradict the fundamental principles of magnetic hydrodynamics and heat transfer.

Thus, the developed mathematical model can be considered acceptable for solving applied problems, and the presented calculation results can be used to further improve the energy and design parameters of multiple-electrode arc furnaces.

## 4. Conclusions

Based on the numerical study of convective heat transfer in the proposed six-electrode furnace supplied with direct current, it was shown that the structure of flows in the furnace was multivortex. Relying on the obtained distribution of the local Richardson number, it was shown that the electric vortex convection dominated near the bath axis and the heat-gravitational convection dominated near the side walls due to the uneven heating of the melt.

The influence of heat-gravitational convection caused a decrease in the width of the main electric vortex flow and formation of an additional flow near the side walls of the furnace, while the velocity of the downward flow dropped down under the electric arcs, and the velocity of the upward flows rose in the area of the bottom electrodes. On average, the indicated change in velocities was 15–25%.

It was shown that, in the areas of the electric arcs and the bottom electrodes, azimuthal flows were formed as three pairs of vortices, which were largely due to the presence of azimuthal electromagnetic forces. The effect of the heat-gravitational convection mainly manifested itself in the area of the bottom electrodes, leading to an increase in the azimuthal velocity and a shift of the centers of the vortices to the bath center.

Based on the analysis of the values of the volume of stagnant zones in the bath (3.8%), the proposed furnace provides acceptable conditions for mixing the melt in the bath of the furnace. Optimization of mixing conditions and evaluation of the economic efficiency of the proposed technical solutions will be considered in future works.

**Author Contributions:** A.K. creating an experimental settig and conducting experimental research on it. V.B. (Vyacheslav Bilousov)—formulation of a mathematical model and a computational algorithm, as well as checking the model for stability and convergence. E.B.—experimental data processing. Comparison of the results of numerical and laboratory experiments. V.B. (Vitaliy Bondarenko)—development of a software package based on the basis of a formulated mathematical model.

**Conflicts of Interest:** The authors declare no conflict of interest.

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
