# Peer review of "The Peculiarities of Convective Heat Transfer in Melt of a Multiple-Electrode Arc Furnace"

_metals, doi:10.3390/met9111174_

Round 1
Reviewer 1 Report
Dear Authors,
The work is interesting. The validation of the results was carried out on the indium-gallium-tin alloy, the temperature was 283 K. The obtained results are valuable, however the assumed of validation conditions differ significantly from the simulation conditions. This should be explained in detail at work. You write at work: The constant parameters of the model, determined by the molten steel properties and the turbulence model constants, are shown in Table 1...
The analysis of the current state of knowledge on this subject is based on publications to which there is no access. For example, works 5, 6, 9, 10.
Minor editing errors. For example, line 182. Double dot.
Yours faithfully
reviewer
Author Response
Point 1. The work is interesting. The validation of the results was carried out on the indium-gallium-tin alloy, the temperature was 283 K. The obtained results are valuable, however the assumed of validation conditions differ significantly from the simulation conditions. This should be explained in detail at work.
Response 1: The article substantiates the conditions of model verification in laboratory installations using low-temperature alloys.
Point 2. You write at work: The constant parameters of the model, determined by the molten steel properties and the turbulence model constants, are shown in Table 1...
Response 2: The phrase in the article is adjusted and finalized.
Point 3. The analysis of the current state of knowledge on this subject is based on publications to which there is no access. For example, works 5, 6, 9, 10.
Response 3: In the list of references to these works added online access addresses
Point 4. Minor editing errors. For example, line 182. Double dot.
Response4: corrected.

Reviewer 2 Report
The paper describes the outcome of a study based on numerical modeling, which investigates magnetic hydrodynamic and thermal processes occurring in the melt of an electric arc furnace equipped with 3 roof arches and 3 bottom electrodes.
The topic of the paper is interesting and fits the aims and scope of the journal. However the paper overall is not well written and shows major flaws which prevent its publication in the present form.
Firstly, the abstract does not introduce the overall industrial problem, which will be overcome by means of the work which is described in the paper. I.e. it is not evident at the abstract stage what it the overall purpose and practical use of the study.
The same weakness, although at a lower level, is also present in the introduction. Moreover, in this chapter, the analysis of the state of the art is totally missing. As a consequence, the authors also do not properly highlight the elements of novelty of the described research activity.
In Section 2, the model is described through its basic equations, while constants and boundary conditions are proposed in the first two tables. However, insufficient explanations and justifications for all the made assumptions are provided here.
In Section 3 the outcomes of the simulations and the interpretation of such results are satisfactorily discussed. However, the validation of the model by means of experimental data coming from 3 plants (a real one and two pilot plants) is only superficially treated. The following sentence:
“A satisfactory qualitative and quantitative coincidence of the results of the flow velocity and temperature calculation was pointed out……“
Is insufficiently justified and poorly supported by the provided data. An “average deviation of 15% between simulated and real data” is mentioned, but it is not clear on which data it has been calculated. Moreover the source of the experimental data shown in Figure 10 is not mentioned and Figure 10 alone does not provide enough evidence of the mentioned good performance of the model.
Author Response
Point 1. Firstly, the abstract does not introduce the overall industrial problem, which will be overcome by means of the work which is described in the paper. I.e. it is not evident at the abstract stage what it the overall purpose and practical use of the study.
Response 1: Abstract redone. Added common purpose and practical use.
Point 2. The same weakness, although at a lower level, is also present in the introduction. Moreover, in this chapter, the analysis of the state of the art is totally missing. As a consequence, the authors also do not properly highlight the elements of novelty of the described research activity.
Response 2: Introduction revised and prior art analysis added.
Point 3. In Section 2, the model is described through its basic equations, while constants and boundary conditions are proposed in the first two tables. However, insufficient explanations and justifications for all the made assumptions are provided here.
Response 3: The article is supplemented by relevant justifications
Point 4. In Section 3 the outcomes of the simulations and the interpretation of such results are satisfactorily discussed. However, the validation of the model by means of experimental data coming from 3 plants (a real one and two pilot plants) is only superficially treated. The following sentence:
“A satisfactory qualitative and quantitative coincidence of the results of the flow velocity and temperature calculation was pointed out……“
Is insufficiently justified and poorly supported by the provided data. An “average deviation of 15% between simulated and real data” is mentioned, but it is not clear on which data it has been calculated. Moreover the source of the experimental data shown in Figure 10 is not mentioned and Figure 10 alone does not provide enough evidence of the mentioned good performance of the model.
Response 4: The article is supplemented with relevant data and explanations to verify the model.

Round 2
Reviewer 1 Report
The Authors replied exhaustively to my comments. I accept these answers.
The article body has been improved. The changes made to the article increase its quality.
In my opinion, the work is suitable for publication in the journal Metals.
Yours faithfully
reviewer
Author Response
Dear Reviewer,
We thank the referee for the fruitful work that significantly improved the article.
Reviewer 2 Report
The paper describes the outcome of a study based on numerical modeling, which investigates magnetic hydrodynamic and thermal processes occurring in the melt of an electric arc furnace equipped with 3 roof arches and 3 bottom electrodes.
The topic of the paper is interesting and fits the aims and scope of the journal.
The authors substantially improved the paper according to the observations of the reviewers and the paper was drastically improved. However, in the current version, still a few drawbacks are present, which need to be eliminated before publication.
At the end of the Introduction, the elements of novelty of the proposed approach should be highlighted in a more formal and systematical way. Please avoid colloquial sentences, such as
“In our opinion, this method of energy supply will allow you to effectively control the eddy currents and heat fluxes in the melt during the melting process.”
The authors must clearly and scientifically state what is novel and strong in the approach they propose.
In Section 2, the following statement has been introduced in order to justify some simplifications:
“…which in our opinion is quite acceptable for the final melting period”.
This statement does not provide any additional information with respect to the rest of the text. The authors must explain WHY the provided assumption can be considered acceptable.
MINOR REMARKS
On Page 4, row 127: at the beginning of the sentence a “T” is missing.
On Page 5, First row of Table 1: please replace “Unit of Measure with” “Unit”
Author Response
Point 1. At the end of the Introduction, the elements of novelty of the proposed approach should be highlighted in a more formal and systematical way. Please avoid colloquial sentences, such as
“In our opinion, this method of energy supply will allow you to effectively control the eddy currents and heat fluxes in the melt during the melting process.”
The authors must clearly and scientifically state what is novel and strong in the approach they propose.
Response 1: This offer has been revised and supplemented.
Point 2. In Section 2, the following statement has been introduced in order to justify some simplifications:
“…which in our opinion is quite acceptable for the final melting period”.
This statement does not provide any additional information with respect to the rest of the text. The authors must explain WHY the provided assumption can be considered acceptable.
Response 2: This offer has been revised and supplemented.
Point 3. MINOR REMARKS
On Page 4, row 127: at the beginning of the sentence a “T” is missing.
On Page 5, First row of Table 1: please replace “Unit of Measure with” “Unit”
Response 3: corrected